# Molecular Characterization of Cephalosporin and Fluoroquinolone Resistant *Salmonella* Choleraesuis Isolated from Patients with Systemic Salmonellosis in Thailand

**DOI:** 10.3390/antibiotics10070844

**Published:** 2021-07-12

**Authors:** Pichapak Sriyapai, Chaiwat Pulsrikarn, Kosum Chansiri, Arin Nyamniyom, Thayat Sriyapai

**Affiliations:** 1Department of Microbiology, Faculty of Science, Srinakharinwirot University, Bangkok 10110, Thailand; peechapack@g.swu.ac.th; 2WHO International Salmonella and Shigella Center, National Institute of Health, Nonthaburi 11000, Thailand; chaiwat.p@dmsc.mail.go.th; 3Department of Biochemistry, Faculty of Medicine, Srinakharinwirot University, Bangkok 10110, Thailand; kosum@g.swu.ac.th; 4Faculty of Environmental Culture and Ecotourism, Srinakharinwirot University, Bangkok 10110, Thailand; arin@g.swu.ac.th

**Keywords:** antimicrobial resistance, cephalosporin, extended-spectrum β-lactamase, fluoroquinolone, *Salmonella* Choleraesuis

## Abstract

The antimicrobial resistance of nontyphoidal *Salmonella* has become a major clinical and public health problem. Southeast Asia has a high level of multidrug-resistant *Salmonella* and isolates resistant to both fluoroquinolone and third-generation cephalosporins. The incidence of co-resistance to both drug classes is a serious therapeutic problem in Thailand. The aim of this study was to determine the antimicrobial resistance patterns, antimicrobial resistance genes and genotypic relatedness of third-generation cephalosporins and/or fluoroquinolone-resistant *Salmonella* Choleraesuis isolated from patients with systemic salmonellosis in Thailand. Antimicrobial susceptibility testing was performed using the agar disk diffusion method, and ESBL production was detected by the combination disc method. A molecular evaluation of *S.* Choleraesuis isolates was performed using PCR and DNA sequencing. Then, a genotypic relatedness study of *S.* Choleraesuis was performed by pulse field gel electrophoresis. All 62 cefotaxime-resistant *S.* Choleraesuis isolates obtained from 61 clinical specimens were multidrug resistant. Forty-four isolates (44/62, 71.0%) were positive for ESBL phenotypes. Based on the PCR sequencing, 21, 1, 13, 23, 20 and 6 ESBL-producing isolates harboured the ESBL genes *bla*_CTX-M-14_, *bla*_CTX-M-15_, *bla*_CTX-M-55_, *bla*_CMY-2_, *bla*_ACC-1_ and *bla*_TEM-1_, respectively. This study also found that nine (9/62, 14.5%) isolates exhibited co-resistance to ciprofloxacin and cefotaxime. All of the co-resistant isolates harboured at least one PMQR gene. The *qnr* genes and the *aac(6′)-Ib-cr* gene were the most prevalent genes detected. The QRDR mutation, including the *gyrA* (D87Y and D87G) and *parC* (T57S) genes, was also detected. PFGE patterns revealed a high degree of clonal diversity among the ESBL-producing isolates.

## 1. Introduction

*Salmonella* is one of the most common types of bacteria; it is a human pathogen that causes ‘Salmonellosis’ and some types of gastroenteritis, including self-limiting enterocolitis, bacteremia, meningitis, and osteomyelitis [1]. More than 2600 different serovars of *S*. *enterica* have been identified. The microorganisms are transmitted by contaminated food or inadequate food hygiene and the fecal–oral route. Southeast Asia has a high level of multidrug-resistant nontyphoidal *Salmonella* and isolates resistant to both fluoroquinolone and third-generation cephalosporins [2,3]. Compared to previous susceptibility patterns among isolates from Southeast Asia, current nontyphoidal *Salmonella* infections in humans in Thailand are more resistant to fluoroquinolone and cephalosporins [4]. The incidence of co-resistance to both drugs classes is a serious therapeutic problem in Thailand.

The production of extended-spectrum β-lactamases (ESBLs) is an important resistance mechanism to extended-spectrum cephalosporins (ESCs) in nontyphoidal *Salmonella* [5]. *Salmonella* spp. have been found to express a wide variety of ESBL genes, including *bla*_TEM_, *bla*_CTX-M_, *bla*_SHV_, *bla*_ACC_, and *bla*_CMY_ [5]. A previous study reported that most ESBLs are coded by plasmids, some of which also carry genes conferring resistance to fluoroquinolones [6]. To date, several plasmid-mediated quinolone resistance (PMQR) mechanisms have been identified: *qnr* gene families encode Qnr protection proteins, *aac(6**′)-Ib-cr* gene encodes variant of an aminoglycoside acetyltransferase, and *oqxA* and *oqxAB* genes encode a multidrug efflux pump [7]. In addition, the mechanisms of quinolone resistance are chromosomal mutations in the quinolone resistance-determining regions (QRDRs) and consist of either the modification of the quinolone targets with changes in the genes encoding DNA gyrase (*gyrA* and *gyrB*) and/or topoisomerase IV (*parC* and *parE*) [8]. A high level of drug resistance arises due to alterations in amino acids on each subunit.

In Thailand, the number of patients who are infected with *Salmonella enterica* serovar Choleraesuis has been increasing. The data show that the prevalence increased from 1.5% (*n* = 87) in 1994 to 9.2% (*n* = 190) in 2006. People between the ages of 6 and 40 years in Thailand have the highest risk of infection [4]. In 2007, an increased number of *S.* Choleraesuis isolates from Thailand were fluoroquinolone- and ceftriaxone-resistant [9]. Lee et al. [10] found that isolates from Thailand and Taiwan demonstrated an alarmingly high frequency of resistance to fluoroquinolones and third-generation cephalosporins. A recent Thai study found that the rate of ESC resistance for *S.* Choleraesuis isolated from bacteremic patients increased from 66.7% in 2010 to 94.1% in 2014 [11]. However, few studies have reported data on antimicrobial resistance to third-generation cephalosporins and fluoroquinolones in *S.* Choleraesuis in Thailand. Therefore, this study aimed to evaluate the antimicrobial resistance patterns, antimicrobial resistance genes and genotypic relatedness of 62 *S.* Choleraesuis isolates collected from patients with systemic salmonellosis in Thailand during 2010–2015.

## 2. Results and Discussions

### 2.1. Antimicrobial Susceptibility and ESBL-Producing S. Choleraesuis Isolates

A total of 62 *S.* Choleraesuis isolates were randomly collected from 61 clinical specimens and used in this study. The majority of *S.* Choleraesuis isolates in this study were recovered from blood specimens which accounted for 83.87% of all strains. According to data obtained from Thailand, *S.* Weltevreden, *S.* Stanley, *S.* Anatum, and *S.* Rissen were the most common serovars in patient stool samples. Conversely, *S.* Choleraesuis, *S.* Enteritidis, *S.* Typhimurium, and *S.* Typhi were observed in blood samples [12]. In a previous study in Thailand during 2005, 135 nontyphoidal *Salmonella* isolated from blood in Siriraj Hospital showed that *Salmonella* group C (47%) was the most common serogroup and 17.8%, 14.1%, 17.8% and 5.8% of isolates were resistant to ceftriaxone, ceftazidime cefotaxime and ciprofloxacin, respectively [13]. Furthermore, *S.* Choleraesuis was the most common serovar (49.3%) in 414 nontyphoid *Salmonella* isolates from bacteremic patients in Thailand between 2005 and 2016 [11].

The antimicrobial susceptibility of 62 *S.* Choleraesuis isolates from patients with systemic infections was evaluated (Table 1). Multidrug resistance (resistance to at least three different classes of antimicrobials) was observed in all isolates. All *S.* Choleraesuis isolates showed reduced susceptibility to 18 antibiotics except for norfloxacin. All isolates were resistant to ampicillin, tetracycline, nalidixic acid and third-generation cephalosporin drugs (ceftriaxone, cefotaxime, ceftriaxone and cefalothin). The susceptibility to antimicrobials was as follows: cefpodoxime (98.4%), ceftiofur (96.8%), chloramphenicol (83.9%), cefuroxime (80.6%), streptomycin (64.5%), ceftazidime (62.9%), aztreonam (43.5%), amoxicillin/clavulanate (43.4%), cefoxitin (40.3%), ciprofloxacin (14.6%), trimethoprim-sulfamethoxazole (11.3%) and cefepime (8.1%). Forty-four *S.* Choleraesuis (44/62 isolates, 71.1%) isolates were positive for ESBL phenotypes. All ESBL-producing isolates were resistant to ampicillin, and 100% were cross-resistant to cefotaxime and cefpodoxime. In total, 18.7% (12/62 isolates) were resistant to all nine cephalosporin antimicrobials (cefepime, cefoxitin, ceftazidime, cefotaxime, cefpodoxime, cephalothin, cefuroxime, ceftriaxone and ceftiofur). Additionally, 62 isolates (100%) and 9 isolates (14.6%) were resistant to nalidixic acid and ciprofloxacin, respectively. However, this study also found that nine isolates exhibited co-resistance to ciprofloxacin with cefotaxime, ceftriaxone and cefalothin.

Different antimicrobial resistance patterns were found amongst the ESBL (*n* = 44) and non-ESBL (*n* = 18) isolates. There was a difference between the level of antimicrobial resistance in the ESBL-positive and ESBL-negative isolates. A total of 30 different resistance patterns were observed amongst the ESBL and non-ESBL producing *S.* Choleraesuis isolates, which were multidrug-resistant to 9 to 16 antimicrobial agents (Table 2). The most common resistance patterns were AMP-AMC-NA-CHL-S-TET-KF-FOX-CXM-CAZ-EFT-CRO-CPD-CTX (four ESBL isolates and four non-ESBL isolates, 12.9%) and AMP-AMC-NA-ATM-CHL-S-TET-KF-FOX-CXM-CAZ-EFT-CRO-CPD-CTX (two ESBL isolates and six non-ESBL isolates, 12.9%), followed by AMP-NA-CHL-TET-KF-CXM-EFT-CRO-CPD-CTX (six ESBL isolates, 9.7%) and AMP-NA-CHL-S-TET-KF-CXM-EFT-CRO-CPD-CTX (four ESBL isolates, 6.4%).

To date, only a few reports from Taiwan and Thailand have described resistance to third-generation cephalosporins and fluoroquinolones in *S.* Choleraesuis [10,14,15]. Our data indicate that the rates of antimicrobial resistance against fluoroquinolones and third-generation cephalosporin in *S.* Choleraesuis are increasing when compared to those previously reported in Thailand. A recent study of 54 *S.* Choleraesuis isolates reports that approximately 60% of the study isolates were nalidixic acid-resistant and 15% were ceftriaxone-resistant [9]. Furthermore, a high rate of ceftriaxone (58.3%) and ciprofloxacin (19.6%) resistance was reported in *S.* Choleraesuis isolated from bacteremia patients in Thailand during 2005–2007 and 2012–2016 [11].

### 2.2. Characterization of the ESBL, QRDR and PMQR Genes

This study provides a description of the diversity of β-lactamase-resistant gene patterns in *S.* Choleraesuis isolates from patients with systemic infection in Thailand. The characteristics of plasmid-mediated ESBL-producing *S.* Choleraesuis isolates are shown in Figure 1. All ESBL-producing isolates have 13 patterns of β-lactamase-resistant genes. Based on PCR sequencing, 35, 23, 20 and 6 ESBL-producing isolates harboured the ESBL genes encoding CTX-M, CMY-2, ACC and TEM-1, respectively. Within the CTX-M family, the CTX-M-9 group is more prevalent than the CTX-M-1 group. The CTX-M-1 and CTX-M-9 groups were represented by CTX-M-55 (13 isolates) or CTX-M-15 (1 isolates) and CTX-M-14 (21 isolates), respectively. The *bla*_OXA_ and *bla*_SHV_ genes were also targeted, but there were no positive results. The occurrence of ESBL phenotypes and genotypes in this study was similar to those of other studies in Taiwan and Thailand [16,17]. The *bla*_CTX-M-14_ and *bla*_CMY-2_ genes were found in extended-spectrum cephalosporinase-producing *S.* Choleraesuis isolates recovered from Thai patients [17]. The *bla*_CMY-2_ gene for ampicillin resistance was also reported in ESC-resistant *S.* Choleraesuis strains isolated in Taiwan in 2006 [14] and Thailand during 2005–2007 [11]. A previous paper has reported that plasmids harbouring the *bla*_CTX-M-14_, *bla*_CTX-M-15_ and *bla*_CTX-M-55_ genes were found in *S.* Typhimurium isolates from paediatric patients in China [18]. To our knowledge, there was only one previous article that reported the dissemination of CTX-M-55-producing *S.* Choleraesuis isolates from ESC-resistant *S.* Choleraesuis clinical isolates in Thailand during 2012–2016 [11]. In the past, plasmids carrying *bla*_CTX-M-55_ and *bla*_CTX-M-15_ have been found in ESBL-producing *E. coli* and *Klebsiella* in China and Thailand [19,20]. The molecular characterization showed that *bla*_CTX-M-55_ is a derivative of *bla*_CTX-M-15_, in which the Ala in position 77 is substituted with a Val. With regard to antibiotic resistance, the MIC ranges for cefotaxime, ceftazidime, ceftriaxone and cefpodoxime of isolates carrying the *bla*_CTX-M-55_ and *bla*_CTX-M-15_ genes were significantly higher than those of isolates carrying *bla*_CTX-M-14_. *The bla*_CTX-M-15_ and *bla*_CTX-M-55_ are the main genes that cause ceftazidime resistance [19]. Ceftazidime-hydrolysing CTX-M-15 and CTX-M-55 enzymes possess a Gly substitution in position 240, which may be associated with a reduced susceptibility to ceftazidime [21]. The data suggest that ESBL-producing isolates have emerged in Thailand on several plasmids and in multiple clones of *S.* Choleraesuis. The clonal spread of CTX-M-mediated resistance may be due to horizontal transfer, but this study did not perform gene or plasmid transfer experiments on our ESBL-producing isolates. However, an epidemiological study of plasmid transfer between bacteria in the *Enterobacteriaceae* family has been reported previously [19,20]. In intensive health-care hospital settings, the development and spread of ESBL genes has most likely been caused or at least facilitated by the overuse of antibiotic drugs [22].

In addition, 20.5% (9/44 isolates) of the ESBL-producing isolates were also co-resistant to ciprofloxacin (Table 3). All co-resistant isolates displayed a QRDR mutation in the *gyrA* (D87Y and D87G) and *parC* (T57S) genes. The most frequently observed point mutations of the *gyrA* gene in fluoroquinolone-resistant *Salmonella* are the amino acid changes at serine-83 to phenylalanine, tyrosine, or alanine, or at aspartic acid-87 to glycine, asparagine, or tyrosine [23]. Mutations in *parC* genes have been found in fluoroquinolone-resistant *S.* Choleraesuis [14]. Of the nine isolates, seven patterns of PMQR genes were identified, all of the co-resistant isolates harboured at least one PMQR gene; the *qnr* genes (*qnrA* or/and *qnrB* or/and *qnrS*) and the *aac(6′)-Ib-cr* gene were the most prevalent resistance genes detected. Based on the MICs of the tested cephalosporins, the MIC values of co-resistant isolates for cefotaxime, ceftazidime, ceftriaxone, and cefpodoxime ranged from 64 to ≥256 μg/mL, 1 to 48 μg/mL, ≥256 μg/mL, and 32 to ≥256 μg/mL, respectively. In addition, four strains isolated from the blood samples showed high MIC levels for ciprofloxacin (2 μg/mL). Only one strain in a 40-year-old patient, SH575/12, showed high MIC levels for both ciprofloxaxin (MIC 2 μg/mL) and cephalosporins (cefotaxime ≥ 256 μg/mL, ceftazidime = 48 μg/mL, ceftriaxone ≥ 256 μg/mL and cefpodoxime ≥ 256 μg/mL). This strain contained two types of ESBL genes (*bla*_TEM-1_ and *bla*_CTX-M-55_), four types of PMQR genes (*qnrA*, *qnrB*, *qnrS* and *aac(6′)-lb-cr*) and presented a mutation in both *gyrA* and *parC*. However, an earlier study demonstrated fluoroquinolone and ESC resistance in *S.* Choleraesuis isolated from bacteremic patients in Thailand in 2003–2005 [9]. In this study, four co-resistant isolates obtained from three patients with a systemic infection showed a higher MIC for ciprofloxacin (2 μg/mL) than previously reported, which reduced the susceptibility to ciprofloxacin to ≥0.125 μg/mL [9].

### 2.3. PFGE Analysis

After digestion by the *Xba*I enzyme, the genetic relatedness of ESBL-producing *S.* Choleraesuis isolates was evaluated by PFGE. The dendrograms in Figure 1 demonstrate the banding patterns and the genetic relatedness of the isolates evaluated by PFGE. Multidrug antimicrobial resistance patterns and ESBL genes were also reported. Thirty-one fingerprint-patterns generated by PFGE were identified amongst the strains recovered from various locations at different times and source isolation, providing information on associations amongst the ESBL-producing strains and evidence of the diversity of ESBL genes harboured on plasmids in Thailand. Forty-four isolates were clustered into six predominant PFGE pulsotypes with ≥2 indistinguishable isolates, which showed a close relationship (Dice correlation coefficient of 95%) based on Tenover criteria [24]. However, this study represented a high diversity of PFGE fingerprint patterns of ESBL-producing *S.* Choleraesuis isolates in Thailand. The PFGE and antimicrobial resistance patterns revealed a high degree of clonal diversity amongst the 44 ESBL-producing isolates. All of the analyses indicate that multiple clones and multiple resistance genes on the plasmids are responsible for the extended-spectrum cephalosporin resistance amongst the *S.* Choleraesuis isolates obtained from patients in Thailand. Similarly, Sirichote et al. [16] reported multiple clones and multiple ESBL resistance genes of *S.* Choleraesuis isolates obtained from patients in Thailand and one patient in Denmark. In addition, 16 unique *Xba*I PFGE patterns of 22 *S.* Choleraesuis isolates were obtained.

## 3. Materials and Methods

### 3.1. Data Source and Salmonella Strains

Pure cultures of cefotaxime-resistant *S.* Choleraesuis were obtained from the WHO National *Salmonella* and *Shigella* Center in Thailand, which were collected from *S.* Choleraesuis isolated on all regions of Thailand. They were selected isolates from patients with systemic salmonellosis in all regions of Thailand during 2010–2015. A total of 62 cefotaxime-resistant *S.* Choleraesuis isolates used in this study were isolated from 61 clinical specimens. *S.* Choleraesuis isolates were obtained from blood (*n* = 53) and other sources (knee joint aspiration (*n* = 3), pus (*n* = 2), stool (*n* = 2), urine (*n* = 1) and thigh swabs (*n* = 1)). The stock culture of *S.* Choleraesuis isolates were re-subcultured to check the purity and were confirmed serovar according to the Kauffmann–White antigen schema [25]. The colonies of *S.* Choleraesuis on xylose lysine deoxycholate agar (XLD agar) appeared colourless, without a black centre. The biochemical test, lysine indole motility (LIM) medium and triple sugar ion (TSI) agar were used for the preliminary identification of *S.* Choleraesuis. The LIM results for *S.* Choleraesuis revealed positive results for lysine decarboxylase and motility, but negative results for indole. The TSI results were presented as alkaline (red) slant and acid butt (yellow), gas positive, and hydrogen sulphide negative.

### 3.2. Antimicrobial Susceptibility Testing

*S.* Choleraesuis isolates were tested for susceptibility to antimicrobial agents using the agar disk diffusion method (Kirby Bauer test) following the guidelines and criteria of the Clinical and Laboratory Standards Institute (CLSI) [26]. The antibiotics tested (Oxoid, England) were ampicillin (10 µg), amoxicillin–calvulanic acid (20/10 µg), aztreonam (30 µg), cefepime (30 µg), ceftazidime (30 µg), cefoxitin (30 µg), cefotaxime (30 µg), cefpodoxime (10 µg), cephalothin (30 µg), cefuroxime (30 µg), ceftriaxone (30 µg), ceftiofur (30 µg), ciprofloxacin (5 µg), chloramphenicol (30 µg), nalidixic acid (30 µg), norfloxacin (10 µg), streptomycin (10 µg), tetracycline (30 µg), and trimethoprim/sulfamethoxazole (25 µg). Furthermore, cefotaxime-resistant *S.* Choleraesuis isolates were screened for ESBL producers by the combination disk method (CLSI, 2008) [12]. The MICs for ciprofloxacin (32–0.002 µg/mL), cefotaxime (256-0.016 µg/mL), ceftazidime (256-0.016 µg/mL), ceftriaxone (256-0.016 µg/mL) and cefpodoxime (256-0.016 µg/mL) were determined using the Liofilchem MIC test strip (Liofilchem, Roseto degli Abruzzi, Italy), according to the manufacturer’s instructions. *Escherichia coli* ATCC 25922 was used as the quality control strain for antimicrobial susceptibility testing.

### 3.3. Characterization of ESBL, QRDR and PMQR Genes

DNA was extracted by the Wizard^®^ Genomic DNA purification kit (Promega, Madison, WI, USA). All primers used for this study are listed in Table 1. The Ex Taq DNA polymerase (Takara Bio Inc., Japan) was used for the detection of ESBL, QRDR and PMQR genes. The ESBL genes, including *bla*_TEM_, *bla*_SHV_, *bla*_CTX-M_, *bla*_ACC_, and *bla*_CMY-2_, were amplified by PCR [27,28,29]. The amplification conditions for the *bla*_TEM_, *bla*_SHV_, *bla*_CTX-M_, *bla*_ACC_, and *bla*_CMY-2_ genes were standardised as follows: an initial denaturation at 95 °C for 5 min, followed by 30 cycles of denaturation at 98 °C for 10 s, annealing at 55 °C for 1 min and DNA extension at 72 °C for 1.30 min, with a final extension at 72 °C for 10 min. For confirmation, the CTX group was amplified and sequenced by the *bla*_CTX-M-1_ group, the *bla*_CTX-M-2_ group, the *bla*_CTX-M-8/25_ group, the *bla*_CTX-M-9_ group and the *bla*_CTX-M-15_ primer [30,31]. The PCR program consisted of an initial denaturation at 95 °C for 5 min, followed by 30 cycles of denaturation at 98 °C for 10 s, annealing at 55 °C for the *bla*_CTX-M-1_, *bla*_CTX-M-2_, *bla*_CTX-M-9_ groups and *bla*_CTX-M-8/25_ groups, and 50 °C for *bla*_CTX-M-15_ for 1 min with a DNA extension at 72 °C for 1.30 min, followed by a final extension at 72 °C for 10 min. The presence of QRDR mutations in *gyrA* and *parC* genes and screening for PMQR, including the *qnrA*, *qnrB*, *qnrS*, and *aac(6**′)-lb-cr* genes, were amplified by PCR [32,33,34]. The amplification conditions for all the PCRs were as follows: an initial denaturation at 95 °C for 5 min, followed by 30 cycles of denaturation at 98 °C for 10 s, annealing at 55 °C (*parC*, *qnrA*, *qnrS*, and *aac(6′)-lb-cr*) and 50 °C (*gyrA* and *qnrB*) for 1 min and DNA extension at 72 °C for 1.30 min, with a final extension at 72 °C for 10 min. All PCR products were purified for sequencing using nucleospin gel extraction (Macherey-Nagel, GmbH & Co. KG, Düren, Germany) and were ran using an automatic sequencer at the Macrogen Inc. (Seoul, Korea). The nucleotide sequence analyses were compared to nucleotide database sequences using the megablast search and to published protein databases using Blastx from the National Center for Biotechnology Information website (https://blast.ncbi.nlm.nih.gov/Blast.cgi, accessed on 21 January 2021).

### 3.4. Pulse-Field gel Electrophoresis (PFGE)

All ESBL-producing isolates included in this study were analysed for epidemiological relatedness by PFGE using *Xba*I according to the Centers for Disease Control and Prevention (CDC) PulseNet protocol [35]. *S*. Braenderup H9812 was used as a standard marker. The electrophoresis was performed with a CHEF-DR III instrument using the following condition: one pulse time of 2.2 to 63.8 s for 19 h. The gel was stained with ethidium bromide, and the DNA bands were visualised with a UV transilluminator. The PFGE profile was analysed by BioNumerics software version 3.0 (Bio-Rad). The similarity index was calculated using the Dice correlation coefficient option of the software with a position tolerance of 1% and an optimization of 0.5%. The dendrogram was created using the unweighted-pair group method with average linkages (UPGMA).

## 4. Conclusions

The present study showed a high occurrence of ESBL producers among *S.* Choleraesuis isolated from patients with systemic salmonellosis in Thailand during 2010–2015. In addition, ESBL-producing isolates showed co-resistance to ESCs and fluoroquinolone. The current study describes the diversity of ESBL genes and identifies QRDR mutations and PMQR genes in ESBL-producing *S.* Choleraesuis. The PFGE typing suggests that ESBL-producing *S.* Choleraesuis strains isolated from patients in different regions in Thailand are multiple clones, which may indicate that isolates of this serovar have spread and that resistance has evolved locally among the isolates in Thailand.

## Figures and Tables

**Figure 1 antibiotics-10-00844-f001:**
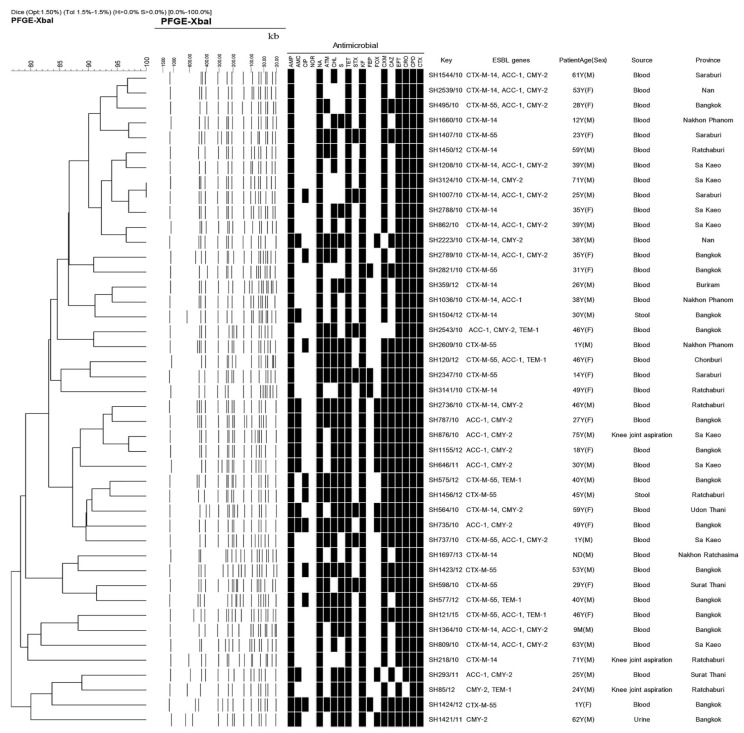
PFGE characteristics and ESBL genes of 44 ESBL-producing *Salmonella*
*Choleraesuis* isolates.

**Table 1 antibiotics-10-00844-t001:** Antimicrobial susceptibilities of 62 *Salmonella*
*Choleraesuis* isolates from clinical specimens.

Antimicrobial	Group	% Resistant (Isolate)	% Intermediate (Isolate)	% Sensitive (Isolate)
AMP	penicillin	100 (62)	0	0
AMC	β-lactam/β-lactam inhibitor combination	43.4 (30)	0	51.6 (32)
ATM	monobactam	43.5 (27)	12.9 (8)	43.5 (27)
FEP	cephalosporin	8.1 (5)	21.0 (13)	71.0 (44)
CAZ	cephalosporin	62.9 (42)	6.4 (4)	25.8 (16)
FOX	cephalosporin	40.3 (25)	4.8 (3)	54.8 (34)
CTX	cephalosporin	100.0 (62)	0	0
CPD	cephalosporin	98.4 (61)	0	0
KF	cephalosporin	100 (62)	0	0
CXM	cephalosporin	80.6 (53)	12.9 (8)	1.6 (1)
CRO	cephalosporin	100.0 (62)	0	0
EFT	cephalosporin	96.8 (60)	3.2 (2)	0
CIP	fluoroquinolone	14.6 (9)	67.7 (42)	17.7 (11)
CHL	phenicol	83.9 (52)	9.7 (6)	6.4 (4)
NA	quinolone	100.0 (62)	0	0
NOR	fluoroquinolone	0.0 (0)	0	100 (62)
S	aminoglycoside	64.5 (43)	19.3 (12)	11.3 (11)
TET	tetracycline	100.0 (62)	0	0
STX	folate Pathway inhibitor	11.3 (7)	1.61 (1)	87.1 (54)

AMP, ampicillin; AMC, amoxicillin + clavulanate; CIP, ciprofloxacin; NA, nalidixic acid; ATM, aztreonam; CHL, chloramphenicol; S, streptomycin; TET, tetracycline; SXT, sulfamethoxazole; KF, cefalothin; FOX, Cefoxitin; FEP, cefepime; CXM, cefuroxime; CAZ, ceftazidime; CTX, cefotaxime; CRO, ceftriaxone; CPD, cefpodoxime; EFT, ceftiofur.

**Table 2 antibiotics-10-00844-t002:** Antimicrobial resistance patterns of ESBL and non-ESBL producing *Salmonella*
*Choleraesuis* isolates from clinical specimens.

Antimicrobial Resistance Patterns	No. of Isolate (%)	
	ESBL	Non-ESBL
**9 antimicrobial resistance** AMP-NA-TET-KF-CXM-EFT-CRO-CPD-CTX	3(4.8%)	-
**10 antimicrobial resistance** AMP-NA-CHL-TET-KF-CXM-EFT-CRO-CPD-CTX AMP-NA-CHL-S-TET-KF-CXM-EFT-CPD-CTX	6(9.6%) 1(1.6%)	- -
**11 antimicrobial resistance** AMP-NA-CHL-S-TET-KF-CXM-EFT-CRO-CPD-CTX AMP-NA-ATM-CHL-TET-STX-KF-EFT-CRO-CPD-CTX AMP-NA-TET-KF-FEP-CXM-CAZ-EFT-CRO-CPD-CTX AMP-NA-ATM-TET-KF-CXM-CAZ-EFT-CRO-CPD-CTX AMP-NA-ATM-CHL-TET-KF-CXM-EFT-CRO-CPD-CTX AMP-AMC-NA-CHL-TET-KF-CXM-EFT-CRO-CPD-CTX AMP-CIP-NA-TET-STX-KF-CXM-EFT-CRO-CPD-CTX AMP-AMC-NA-S-TET-KF-FOX-CAZ-CRO-CPD-CTX	4(6.4%) 1(1.6%) 1(1.6%) 1(1.6%) 1(1.6%) 1(1.6%) 1(1.6%) -	- - - - - - - 1(1.6%)
**12 antimicrobial resistance** AMP-CIP-NA-ATM-CHL-TET-KF-CXM-EFT-CRO-CPD-CTX AMP-AMC-NA-CHL-S-TET-KF-FOX-CAZ-CRO-CPD-CTX AMP-NA-S-TET-KF-FEP-CXM-CAZ-EFT-CRO-CPD-CTX AMP-AMC-NA-CHL-S-TET-KF-CAZ-EFT-CRO-CPD-CTX AMP-AMC-NA-TET-KF-FOX-CXM-CAZ-EFT-CRO-CPD-CTX	1(1.6%) 1(1.6%) 1(1.6%) - -	- - - 2(3.2%) 1(1.6%)
**13 antimicrobial resistance** AMP-NA-ATM-CHL-TET-STX-KF-CXM-CAZ-EFT-CRO-CPD-CTX AMP-NA-ATM-CHL-S-TET-KF-CXM-CAZ-EFT-CRO-CPD-CTX AMP-NA-ATM-S-TET-STX-KF-CXM-CAZ-EFT-CRO-CPD-CTX AMP-CIP-NA-ATM-CHL-S-TET-KF-CXM-EFT-CRO-CPD-CTX AMP-AMC-NA-CHL-S-TET-KF-FOX-CAZ-EFT-CRO-CPD-CTX	2(3.2%) 2(3.2%) 1(1.6%) 1(1.6%) -	- - - - 3(4.8%)
**14 antimicrobial resistance** AMP-AMC-NA-CHL-S-TET-KF-FOX-CXM-CAZ-EFT-CRO-CPD-CTX AMP-CIP-NA-ATM-CHL-S-TET-KF-CXM-CAZ-EFT-CRO-CPD-CTX AMP-AMC-NA-ATM-CHL-S-TET-KF-FOX-CAZ-EFT-CRO-CPD-CTX	4(6.4%) 4(6.4%) 1(1.6%)	4(6.4%) - -
**15 antimicrobial resistance** AMP-AMC-NA-ATM-CHL-S-TET-KF-FOX-CXM-CAZ-EFT-CRO-CPD-CTX AMP-AMC-NA-CHL-S-TET-STX-KF-FOX-CXM-CAZ-EFT-CRO-CPD-CTX AMP-NA-ATM-CHL-S-TET-STX-KF-FEP-CXM-CAZ-EFT-CRO-CPD-CTX AMP-AMC-NA-ATM-CHL-S-TET-KF-FEP-CXM-CAZ-EFT-CRO-CPD-CTX	2(3.2%) 1(1.6%) 1(1.6%) -	6(9.6%) - - 1(1.6%)
**16 antimicrobial resistance** AMP-AMC-CIP-NA-ATM-CHL-S-TET-KF-FOX-CXM-CAZ-EFT-CRO-CPD-CTX AMP-AMC-CIP-NA-ATM-CHL-S-TET-KF-FEP-CXM-CAZ-EFT-CRO-CPD-CTX	1(1.6%) 1(1.6%)	- -

AMP, ampicillin; AMC, amoxicillin + clavulanate; CIP, ciprofloxacin; NA, nalidixic acid; ATM, aztreonam; CHL, chloramphenicol; S, streptomycin; TET, tetracycline; SXT, sulfamethoxazole; KF, cefalothin; FOX, cefoxitin; FEP, cefepime; CXM, cefuroxime; CAZ, ceftazidime; CTX, cefotaxime; CRO, ceftriaxone; CPD, cefpodoxime; EFT, ceftiofur.

**Table 3 antibiotics-10-00844-t003:** Phenotypic and genotypic features of the co-resistant *Salmonella* Choleraesuis isolates.

LabID	MIC CIP (32–0.002µg/mL)	QRDR Mutation in:		PMQR	MIC Cephalosporin (256–0.016 µg/mL)				ESBL Gene
		*gyrA*	*parC*		CTX	CAZ	CRO	CPD	
SH1007	0.25	D87Y	T57S	*qnrA*, *qnrS1*	128	1.5	≥256	96	*bla*_CTX-M-14_, *bla*_ACC-1_, *bla*_CMY-2_
SH2609	2	D87G	T57S	*qnrB*, *qnrS1*, *aac(6′)-lb-cr*	≥256	32	≥256	≥256	*bla* _CTX-M-55_
SH2789	0.5	D87Y	T57S	*qnrA*, *qnrB*, *qnrS1*, *aac(6′)-lb-cr*	64	1	≥256	32	*bla*_CTX-M-14_, *bla*_ACC-1_, *bla*_CMY-2_
SH735	2	D87Y	T57S	*qnrS1*, *aac(6′)-lb-cr*	64	32	≥256	32	*bla*_ACC-1_, *bla*_CMY-2_
SH575	2	D87G	T57S	*qnrA*, *qnrB*, *qnrS1*, *aac(6′)-lb-cr*	≥256	48	≥256	≥256	*bla*_CTX-M-55_, *bla*_TEM-1_
SH1423	0.5	D87G	T57S	*qnrS1*, *qnrB*, *aac(6′)-lb-cr*	≥256	32	≥256	≥256	*bla* _CTX-M-55_
SH577	2	D87G	T57S	*qnrA*, *qnrB*, *qnrS1*	≥256	32	≥256	≥256	*bla*_CTX-M-55_, *bla*_TEM-1_
SH1424	0.12	D87G	T57S	*qnrA*, *qnrB*, *qnrS1*, *aac(6′)-lb-cr*	≥256	48	≥256	≥256	*bla* _CTX-M-15_
SH1456	0.25	D87G	T57S	*qnrA*, *qnrS1*, *aac(6′)-lb-cr*	≥256	32	≥256	≥256	*bla* _CTX-M-55_

CTX, cefotaxime; CAZ, ceftazidime; CRO, ceftriaxone; and CPD, cefpodoxim; D, aspartic acid; G, glycine; S, serine; T, threonine; Y, tyrosine.

## Data Availability

Data is contained within the article.

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
