# Peer review of "Molecular Characterization of Cephalosporin and Fluoroquinolone Resistant Salmonella Choleraesuis Isolated from Patients with Systemic Salmonellosis in Thailand"

_antibiotics, 2021, doi:10.3390/antibiotics10070844_

Round 1
Reviewer 1 Report
Dear authors
The specialized articles in the field of bacterial resistance mechanisms to the existing therapeutic arsenal are very topical. The article is an interesting one. My recommendation is to publish with minor revision:
- One of the objectives proposed by the authors in the present study is to establish a mechanism by which the salmonella isolates studied are resistant to the two classes of antibacterial compounds. In my opinion, what the authors present are rather results without a deep interpretation of them. I recommend the authors to complete the discussions of the results obtained in this respect.
-I also recommend the authors to highlight more clearly what is original in the research presented.
-the article requires a careful verification of the language used.
Reviewer 2 Report
The findings are definitely of interest for the scientific community and worth publishing, but we would like to make the following suggestions for correction or completion:
Lines 15 and 43: please correct “nontypoidal” to “nontyphoidal”. “Salmonella Cholaresuis” should be replaced by “Salmonella enterica serotype Choleraesuis”.
“All cefotaxime-resistant S. Choleraesuis isolates were obtained from 61 clinical specimens were multi-drug resistant.” This sentence needs rephrasing for correctness and clarity.
Lines 61-72: the data cited there are quite old (about 15 years now), it would be useful to cite more contemporary data (e.g. in the last 3-5 years).
Lines 296:298: although NCBI makes available multiple blasting algorithms, they have many default parameters and options, allowing a wide space of customization. The authors should specify if they used the blasting with the default parameters, or otherwise clarify what parameters were used (in the text or in the supplementary materials).
Lines 101-102: this sentence is unclear (what is the main difference between the 100% and the 14.6% subset?
Table 1: the use of AMP abbreviation for amoxicillin is rather unusual, because AMP is usually reserved for ampicillin (whereas for amoxicillin the abbreviation is AMX – see, e.g. http://bsacsurv.org/science/antimicrobials/ ).
Line 183: we would suggest replacing “most likely been caused by” with “most likely been caused or at least facilitated by...”
The materials and methods section should provide minimal information on the software and method/key parameters used in generating the dendrogram.
The discussions should preferably focus a bit on the mechanisms of the genes identified in antibiotic resistance, because for someone not familiar with the genes, the lecture may be devoid of interest, where converting technical information in human understandable language would result in an increase in quality and higher interest of the potential readers for the findings.
Reviewer 3 Report
In this study, the authors characterized 62 isolates of Salmonella serovar Choleraesuis isolated in Thailand. The authors describe the resistance patterns identified following antimicrobial susceptibility testing by the agar disk-diffusion method. They also perform PGFE analysis on the ESBL isolates.
The study is of great interest and the science used is accurate. However, the manuscript requires revision in terms of sentence structure and will improve after removal of many typographical errors. I highly recommend the authors to revise the manuscript using the comments provided.
I also believe the Conclusion section can be re-written to emphasize the findings of this study.
COMMENTS:
There are some inconsistencies in the abbreviations used for the same antimicrobials in Table 1, Table 2 and Figure 1. These need to be corrected. For example, in Table 1 nalidixic acid is represented by NAL but in Table 2 and in Figure 1, it is represented by NA. Same problem with streptomycin (S or STR)
Since the authors are using CLSI guidelines, the units for MIC values should be given as mg/L (and not ug/mL)
Line 15: “nontyphoidal” is incorrectly spelt
Line 18: replace “drugs” with “drug classes”
Line 21: “agar diffusion” should read “agar disk diffusion”
Line 25: remove “were”
Line 30: there should be a full-stop after PMQR (not a comma)
Line 31: should read “The QRDR mutations of gyrA (D87Y and D87G) and parC (T57S) were also detected”. Also, ensure gyrA is spelt correctly.
Line 38: Remove “is one of the most common types of bacteria”. Also, spp. is plural - should use correct grammar.
Line 41: “enterica” should be in italics (species name)
Line 43: The authors use “multi-drug”, but elsewhere they use “multidrug” (without the hyphen) - choose one and be consistent.
Line 43: “nontyphoidal” is incorrectly spelt
Line 43: “Salmonella” should be in italics (genus name)
Line 45: “Salmonella” should be in italics
Line 47: “drugs” should read “drug classes”
Line 49: “Salmonella” should be in italics
Line 50: “Salmonellae” should read “Salmonella spp.”
Line 51: should start with “A previous study …”
Line 52: “fluoroquinolones” in plural
Revise lines 54-55. Genes and proteins are different. The protein is “encoded” by the gene. Same problem on line 58. We cannot say “DNA gyrase genes” - we say “the genes that encode DNA gyrase”.
Line 56: should read “In addition, quinolone resistance is mediated by chromosomal mutations ….”
Line 61: use “Salmonella serovar Choleraesuis” for the first instance
Line 63: specify the unit for ages of 6 and 40 - years.
Line 67: use “third” - to be consistent with the rest of the manuscript
Line 75: “isolated” should read “isolates”
Line 76: use “The majority..”
Line 77: use “.. which accounted for..”
Line 87: use “The antimicrobial susceptibility profiles of ……… were evaluated (Table 1).”
Line 91: remove 100%
Lines 92-96 is a repetition of the contents of Table 1. Can delete this sentence.
Lines 109-111: be consistent with naming of antibiotics here – as currently written, some start with a capital letter (e.g. ciprofloxacin) but some don’t (e.g. amoxicillin) – I suggest starting all with a small letter.
Line 119: In the resistance pattern provided, NA should be NAL and S should be STR. Correct throughout the text.
Line 138: “reported” should read “reports”
Line 149: “the description” should read “a description”
Line 151: remove genes (isolates don’t produce genes)
Line 157: “gene was” should be replaced with “genes were”
Line 166: “… previous article that reported the dissemination of…”
Line 167: “Choleraesuis” should not be in italics
Line 186: remove “including”
Line 193: replace “mutations” with “resistance genes”
Line 194: sentence does not make sense. revise the grammar.
Lines 195-207: replace ug/mL with mg/L
Line 196: remove “that”
Line 205: a space between 3 and patients
Line 206: sentence does not make sense. Revise.
Line 231: “D” is aspartic acid (NOT asparagine!)
Line 243: “Cefotaxime-resistant” should be with a small “c”
Line 247: S. Choleraesuis “isolates”
Lines 243-257: Please revise the English.
Line 288: should read “The presence of QRDR mutations in gyrA and parC …”
Line 309: The first sentence of the conclusion does not make sense. Please revise.
Line 313: “have” should be corrected to “has”
Round 2
Reviewer 1 Report
Dear authors
I am agree with the publication of the article in the current form. Congratulations.